# Skin Adverse Reactions to Novel Messenger RNA Coronavirus Vaccination: A Case Series

**DOI:** 10.3390/diseases9030058

**Published:** 2021-08-27

**Authors:** Maria Francesca Peigottu, Caterina Ferreli, Maria Giovanna Atzori, Laura Atzori

**Affiliations:** 1Dermatology Unit, San Francesco Hospital, 08100 Nuoro, Italy; mariafrap@tiscali.it (M.F.P.); mgatzori.mga@gmail.com (M.G.A.); 2Dermatology Clinic, Department Medical Sciences and Public Health, Cagliari State University, 09124 Cagliari, Italy; ferreli@unica.it

**Keywords:** skin adverse reactions, novel messenger RNA coronavirus vaccine, COVID-19 vaccination, delayed skin reactions

## Abstract

Vaccines are actually the most effective strategy to control the COVID-19 spread and reduce mortality, but adverse reactions can occur. Skin involvement with novel messenger RNA coronavirus vaccines seems frequent but is not completely characterized. A real-world experience in the recent vaccination campaign among health care workers in Sardinia (Italy) is reported. In over a total of 1577 persons vaccinated, 9 cases of skin adverse reactions were observed (0.5%). All reactions have been reported to the Italian Pharmacovigilance Authority. Eight occurred in women (mean age 46 years), and five were physicians and four nurses. All patients had a significant allergology history but not for the known vaccine excipients. After dose one, no injection site reactions were observed, but widespread pruritus (*n* = 3), mild facial erythema (*n* = 1), and maculopapular rash (*n* = 3) occurred in the following 24–48 h in three patients. These three patients were excluded from the second dose. Of the remaining six patients, one developed mild anaphylaxis within the observation period at the vaccination hub and five delayed facial erythematous edema and maculopapular lesions, requiring antihistamines and short-course corticosteroid treatment. Spontaneous reporting is paramount to adjourning vaccination guidance and preventive measures in order to contribute to the development of a safe vaccine strategy. Dermatologist’ expertise might provide better characterization, treatment, and screening of individuals at high risk of skin adverse reactions.

## 1. Introduction

The novel messenger RNA (mRNA) technology vaccines are a crucial part of the worldwide fight against coronavirus disease 2019 (COVID-19), and the Pfizer-BioNTech vaccine has been the first issued in Italy for the health care workers’ vaccination program. Reported adverse effects in clinical trials include minimal local skin reactions, especially following the required second dose of the vaccine, without systemic adverse reactions [1,2]. However, individuals with a history of a severe adverse reaction related to the vaccine and/or other severe allergic reactions (e.g., anaphylaxis) were excluded from pivotal approval studies, and in a registry-based study of 414 cases, a spectrum of cutaneous reactions after mRNA COVID-19 Moderna (83%) and Pfizer (17%) vaccines was reported [3]. Forty-three percent of patients with first-dose reactions experienced second-dose recurrence. No serious adverse events developed in any of the patients, thus not discouraging vaccination.

However, the Centers for Disease Control and Prevention (CDC) recommends to avoid a second dose if a severe or immediate allergic reaction to the first dose of an mRNA COVID-19 vaccine has occurred [4]. Assessment with excipient skin testing in 80 patients reporting an allergic reaction to mRNA COVID-19 vaccine dose one, a sensitization to PEG (*n* = 5) and/or polysorbate 80 (*n* = 12) was documented in a minority of patients, and most of the patients received the second dose safely [5].

The present short case series documents the real-world experience in the recent vaccination campaign among health care workers in Sardinia, a major island in the Mediterranean basin, and all reactions have been reported to the Italian Pharmacovigilance Authority.

## 2. Case Series

Of over a total of 1577 persons vaccinated during the first vaccination round, which ended in January 2021, 9 cases of skin adverse reactions were observed (0.5%). As regards demographics (Table 1), eight occurred in women, the age ranged from 35 to 55 years (mean 46 years), five were physicians, and four were nurses.

All patients had a significant allergology history: atopy in five, penicillin allergy in four, additional drug allergy in three, and contact dermatitis in three patients. However, none of the patients had a documented allergy to substances regarded as potentially predisposing to mRNA vaccines, such as latex, polyethylene glycol, polyoxyl castor oil, or polysorbate. Angioedema and anaphylaxis were reported in three patients. It is noteworthy that the Italian guidelines on the allergology screening for patients with a history of asthma and allergy were released quite after the first dose of the vaccine had been performed [6], and the decision whether to administrate dose two had to be individualized. Priority to the broad vaccination coverage of the health care community was recognized in the midst of the pandemic. It should be considered that Nuoro has been the second city in Sardinia for the number of COVID-19 cases, with a high percentage of health care workers severely affected.

As regards presentation of skin adverse events, and timing, no injection site reactions were observed. After the first dose, two patients did not manifest any reaction, three patients noted late-onset widespread pruritus, and one showed mild transitory facial erythema, not requiring intervention. In the remaining three patients, a maculopapular rash developed 48 h after injection. Allergology consulting excluded administration of the second dose in the latter three patients, while the risk for the other six patients was considered low and the vaccination was completed. One patient experienced glottis edema, within the observation timing at the vaccination hub, receiving epinephrine, but with fast and complete recovery. The other five patients, 24–48 h after injection, developed facial erythematous edema and widespread itching maculo-papular lesions (Figure 1), Mild general malaise, with muscle weakness and migrating arthralgias but not fever, was also reported. Treatment consisted of oral antihistamines and a short course of systemic steroids.

## 3. Discussion

As dermatologists, we are actively committed to supporting the Vaccine Adverse Event Reporting System (VAERS) and enhancing continuous safety monitoring [7]. Severe allergic reactions to vaccines are rare but can be life threatening. The BNT162b2 vaccine is based on lipid nanoparticles and other substances to enable transport of messenger RNA (mRNA) molecules into the cells, which can be potential allergens. In Pfizer-BioNTech COVID-19 vaccine approval clinical trials, injection site pain was frequently reported (84.1% of recipients), followed by swelling (10.5%) and erythema (9.5%) [2]. The McMahon et al. registry documented delayed large local reactions, followed by local injection site reactions and several manifestations that mimicked SARS-CoV-2 infection itself, such as pernio/chilblains, urticarial eruptions, and morbilliform eruptions [3]. Less commonly described reactions included cosmetic filler reactions, zoster, herpes simplex flares, and pityriasis rosea-like reactions. Unfortunately, the study did not report the number of vaccinated individuals among which the 414 patients developed the skin manifestations. Nor did it allow a real evaluation of the differences between the two vaccines, as the relative safety of Pfizer in respect to Moderna reactions might depend on a less extensive use of the Pfizer vaccine in the United States.

Our experience confirms that Pfizer-BioNTech vaccine skin reactions are uncommon (0.5%) but suggests implementing allergology screening and prophylaxis measures, not to undervalue delayed skin reactions. Clinical trials and post-marketing experience focus on anaphylactic reactions that occurred within a 30 min observation window, and patients were treated immediately with complete resolution of symptoms [2,8]. However, only one of our patients developed anaphylaxis while on the vaccination hub. All other patients had delayed reactions. The health care workers involved in this Italian vaccination program were carefully informed and had specific knowledge of adverse reaction occurrence, allowing prompt recognition and treatment, including self-medication, which might be not the case with the general population, alone at home after the vaccination procedure.

Alerting signs after dose one (Table 2) that should be considered are a generalized itching and even slight facial erythematous edema, requiring allergology consulting. A careful allergology personal history appears the most important moment, which should not be limited to the evaluation of known vaccine excipient sensitization. Skin testing before vaccine dose two administration was not predictive in a recent study, with negative skin tests in patients who developed reactions to their 388 mRNA COVID-19 vaccine second dose [5].

In our experience, the personal allergy history was positive for penicillin and other drug reactions, instead of the allergens considered related to the messenger RNA (mRNA) vaccine, such as polyethylene glycol, polyoxyl castor oil, or polysorbate.

Another uncovered issue is the opportunity of a premedication regimen, such as with non-sedating antihistamines for patients with mild symptoms at dose one [3,4,7]. Major health care agencies, including the CDC, discourage preventive administration of antihistamines, because there is no current evidence that they cannot interfere with vaccine effectiveness. However, in a recent experimental setting, it was demonstrated that histamine H1 antagonists, especially desloratadine, could bind angiotensin-converting enzyme 2 (ACE2), blocking interactions with the viral spike (S) glycoprotein, thus preventing the virus’s entrance into cells [9]. Antihistamines seems to work like antibodies. The COVID-19 molecules of mRNA enter the membrane via an endocytosis mechanism and are released into the cytoplasm of the muscle cells, independently from angiotensin-converting enzyme 2 (ACE2). Once inside the cell, mRNA gets translated by a ribosome and induces the release of SARS-CoV-2 spike protein, which can be detected by immune cells in the tissue and induce antibody production. Thus, antihistamines should not interfere with mRNA vaccine entrance and functions. If confirmed, antihistamines are candidates as an effective therapeutic option for COVID-19, and might be considered a safe preventive measure against vaccine reactions in allergic individuals.

The low rate of reactions and favorable course in all the nine patients might be helpful to reassure patients as well as health care providers that the vaccination program is safe and affordable even in allergic patients.

## 4. Conclusions

Careful surveillance over time and spontaneous reporting of mRNA-based COVID-19 vaccine safety are crucial measures to better characterize morphology and timing, as well as predisposing conditions. Dermatologists’ expertise in proper diagnosis will contribute to the definition of the spectrum of skin adverse reactions and provide insights into the possible mechanisms, as well as suggest appropriate screening for high-risk individuals. Our case series does not reach a statistical power, and large epidemiological studies will be hopefully provided by nation-wide registers. No premedication was supplied to the patients, following the Italian recommendation for SARS-CoV-2 at the vaccination hub. So, only the vaccine composition should be associated with the reactions. As regards skin testing, patients with more severe reactions were evaluated by the allergologists and were negative for vaccine excipients and cross-reactive substance sensitization. The fact that the skin reaction in some patients was triggered only by the second dose is a common finding with drugs adverse reactions and was confirmed in recent 388 mRNA COVID-19 vaccine studies. It is for this reason that intradermal testing for a local anesthetic is no more routinely performed, as in several cases, the reaction was negative at testing, while the local injection during following surgery or a dentistry procedure triggered true anaphylaxis.

As the world fights COVID-19, the medical community is gaining more experience in the risk–benefit assessment before vaccination, and guidance should be regularly updated. Avoidance of any hazard is necessary to reassure patients of the need to accept this important preventative measure.

## Figures and Tables

**Figure 1 diseases-09-00058-f001:**
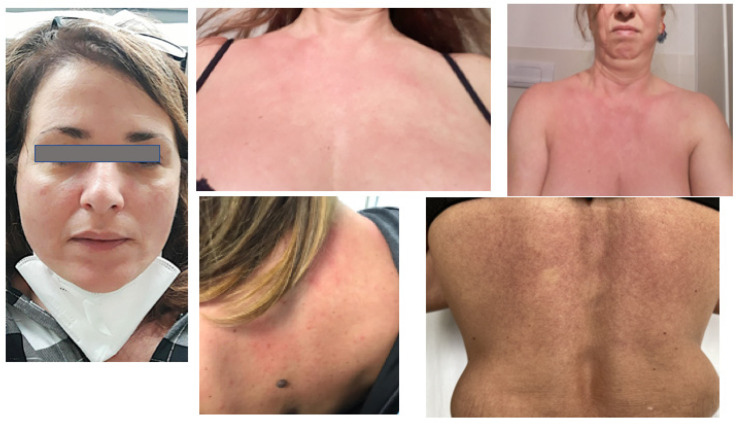
Erythematous edematous and maculopapular eruptions in 4 different patients after Pfizer/BioNTech vaccination.

**Table 1 diseases-09-00058-t001:** Patients’ data and experienced cutaneous adverse reactions to Pfizer-BioNTech vaccination.

Patient Sex and Age	Work	Allergology Anamnesis	1st Dose	2nd Dose	Treatment
Woman, 40 years	Physician	Atopy, anaphylaxis to penicillin and cephalosporin	No reaction	Face erythema and swelling 20 min after injection	Spontaneous remission in 30 min
Woman, 52 years	Nurse	Penicillin allergy, nickel allergy	Widespread itch	Urticarial rash 30 min after injection	Antihistamine + short-course steroids
Woman, 55 years	Nurse	Atopy, cephalosporin allergy, lactose intolerance	Widespread itch and facial erythema 8 h after injection	Erythematous maculopapular eruption 24 h after injection	Antihistamine + short-course steroids
Woman, 35 years	Physician	Atopy	Glottis edema and facial erythema 20 min after injection	Not performed	Antihistamine + short-course steroids
Woman, 51 years	Nurse	Atopy, anaphylaxis	Urticarial rash with arthralgias in the 24 h after injection	Not performed	Short-course steroids
Woman, 62 years	Physician	Previous angioedema (no identified cause)	Widespread maculopapular eruption 24 h after injection	Not performed	Antihistamine
Woman, 37 years	Nurse	Penicillin allergy	Persistent itch, especially on the scalp	Erythematous maculopapular eruption 48 h after injection	Antihistamine + short-course steroids
Woman, 41 years	Physician	Atopy, drug allergy	Widespread itch	Erythematous maculopapular eruption 48 h after injection	Antihistamine
Man, 41 years	Physician	Atopy, drug allergy	No reaction	Erythematous maculopapular eruption 48 h after injection	Antihistamine

**Table 2 diseases-09-00058-t002:** Post-vaccinated dose one reactions and second dose risk assessment.

Clinical Phenotyping	Second Dose Risk	Advertisements
No reactions at first dose, but history of atopy, penicillin, or other drug allergy, contact dermatitis	Low risk	Second dose: instruct the patient about delayed reaction (>4 h) reporting
Immediate skin reactions (<4 h) but not anaphylaxis		
Only subjective symptoms	Low risk	Second dose: instruct the patient about delayed reaction (>4 h) reporting
Widespread itch Facial erythema	Medium risk	Second dose with prolonged post-vaccination observation: support for delayed reactions
Labial–glottis edema	High risk	Allergology evaluation and skin testing for vaccine excipients (i.e., PEG 3350) Decision sharing with supervision physician on second dose vaccination
Delayed skin reactions (>4 h)		
Widespread itch, general malaise, arthralgias Facial erythema Large local reaction Urticarial rash Maculopapular rash	Medium-to-high risk	Allergology evaluation and skin testing for vaccine excipients (i.e., PEG 3350) Decision sharing with supervision physician on second dose vaccination

## Data Availability

The data presented in this study are available on request from the corresponding author.

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
