# Peer review of "Skin Adverse Reactions to Novel Messenger RNA Coronavirus Vaccination: A Case Series"

_diseases, 2021, doi:10.3390/diseases9030058_

Round 1

Reviewer 1 Report

The author reported case series of the skin adverse reactions to mRNA Coronavirus vaccines. Although not common, there were some with generalized skin involvement. There were some issues to be discussed.

  1. The most important thing for observation of skin reaction is if it's part of the ongoing anaphylaxis reaction. If it is, the aggressive treatment for anaphylaxis should be done and the next dose (if feasible) should be avoided. However, if it's just an allergy reaction without life-threatening status, the 2nd dose of vaccine could be administered with symptomatic treatment of the skin reactions. The author should give some clues for the readers how to distinguish the skin reactions that does matter with life-threatening status.
  2. In discussion, the author said that histamine H1 antagonists, especially desloratadine could bind angiotensin-converting enzyme 2 (ACE2), blocking interactions with the viral spike (S) glycoprotein, thus preventing the virus entrance into cells. Does this process interfere the vaccination immune reaction / vaccine effectiveness? If it does, the preventive use of antihistamine should not be recommended for vaccination.

Author Response

Thanks for the consideration and suggestions.

  • The issue of how to distinguish among life-threatening and not worrisome conditions is still a matter of debate and we have followed algorithm suggest by allergology and dermatologist Scientific Society, already reported among references, and in the text. As per your kind request, we propose to add a table (table 2) to highlight such possible clues. We hope this will be helpful in general medical practice.
  • We agree and have reported that at the moment the preventive use of antihistamines is controversial and not recommended by CDC and other Medical Agencies. However, we found interesting the experimental finding of the potential role as therapeutic agents of antihistamines, and included in the discussion. Antihistamines seems to work like antibodies to block virus entrance into the cells. About interference with vaccine efficacy, COVID-19 molecules of messenger RNA (mRNA) enter the cells and are released into the cytoplasm of the muscle cells, independently from angiotensin-converting enzyme 2 (ACE2). Once inside the cell, mRNA gets translated by a ribosome and induce the release a lot of SARS-CoV-2 spike protein, which can be detected by immune cells in the tissue, and induce antibodies production. Thus, antihistamines should not interfere with mRNA vaccine entrance and functions. Of course, our speculations need scientific evidence and are not against current general Health Care Agencies recommendation. We are sorry, presentation might be confusing for the reader. We propose the following changes in the text:  

“Another uncovered issue is the opportunity of a premedication regimen, such as with non-sedating antihistamines for patients with mild symptoms at dose one [3, 4, 7]. Major health care agencies, including CDC discourage preventive administration of antihistamines, because there is no current evidence that they cannot interfere with vaccine effectiveness.  However, in a recent experimental setting, it was demonstrated that histamine H1 antagonists, especially desloratadine could bind angiotensin-converting enzyme 2 (ACE2), blocking interactions with the viral spike (S) glycoprotein, thus preventing the virus entrance into cells [9]. Antihistamines seems to work like antibodies. The COVID-19 molecules of mRNA enter the membrane with a endocytosis mechanism and are released into the cytoplasm of the muscle cells, independently from angiotensin-converting enzyme 2 (ACE2). Once inside the cell, mRNA gets translated by a ribosome and induce the release of SARS-CoV-2 spike protein, which can be detected by immune cells in the tissue, and induce antibodies production. Thus, antihistamines should not interfere with mRNA vaccine entrance and functions.”

Reviewer 2 Report

This is a nice case series about adverse reactions to COVID vaccines in Italy. The data are clearly presneted. Phtodocumentation is of good quality. References are up-to-date.

Author Response

Thanks so much for the kind words and consideration. 

Reviewer 3 Report

The article is not having enough statistical power, it would help to include a bit more case study or sample size. 0.5% adverse reaction is low, also it's not clear why the adverse reaction occurred, was it due to premedication, or was it just the vaccine composition. I like the idea of testing the skin response. It is also not clear why after 1st dose there was no reaction but the 2nd dose triggered an adverse responce. 

Author Response

Of course, we agree with the reviewer that the study does not reach a statistical power, which is however behind the scope of this brief comunication on active pharmacovigilance reporting. Large epidemiological studies will be hopefully provided by nation wide registers. No premedication were supplied to the patients, following the Italian raccomandation for Sars-CoV-2 at vaccination hub. So, only vaccine composition should be associated with the reactions. As regards skin testing,  patients with more severe reactions were evaluated by the allergologists and the negativity for the eccipients and common reported cross-reactive substances sensitization confirmed. The fact that the reaction in some patients was triggered only from the second dose is a common findings with drugs adverse reactions, supporting the role of a privious sensitization phase. It is for this reason that intradermal testing of local anesthetic in no more routinarily performed, as in several cases the reaction was negative at testing and triggered true anaphilaxys while performing the local surgical or dentistry procedure. 

Round 2

Reviewer 3 Report

Can the final response be included in the final conclusion summary of the paper? So the readers are aware.

Author Response

Thanks for the reply. The following sentence has been added to the conclusion section:

"Our case series does not reach statistical power, and large epidemiological studies will be hopefully provided by nationwide registers. No premedication was supplied to the patients, following the Italian recommendation for Sars-CoV-2 at the vaccination hub. So, only vaccine composition should be associated with the reactions. As regards skin testing, patients with more severe reactions were evaluated by the allergologists and were negative for vaccine excipients and cross-reactive substances sensitization. The fact that the skin reaction in some patients was triggered only by the second dose is a common finding with drug adverse reactions and confirmed in recent 388 mRNA COVID-19 vaccine studies. It is for this reason that intradermal testing for local anesthetic is no more routinely performed, as in several cases the reaction was negative at testing, while the local injection during following surgery or dentistry procedure triggered true anaphylaxis."